# The Impact of the COVID-19 Pandemic on Immunization Campaigns and Programs: A Systematic Review

**DOI:** 10.3390/ijerph18030988

**Published:** 2021-01-22

**Authors:** Zohra S. Lassi, Rabia Naseem, Rehana A. Salam, Faareha Siddiqui, Jai K. Das

**Affiliations:** 1Robinson Research Institute, The University of Adelaide, Adelaide, SA 5005, Australia; zohra.lassi@adelaide.edu.au; 2Division of Women and Child Health, The Aga Khan University, Karachi 74500, Pakistan; rabianaseemm@gmail.com (R.N.); rehana.salam@aku.edu (R.A.S.); Faareha.siddiqui@gmail.com (F.S.)

**Keywords:** immunization, vaccination, coronavirus, COVID-19, polio

## Abstract

The COVID-19 pandemic has had an impact on health service delivery, including immunization programs, and this review assesses the impact on vaccine coverage across the globe and identifies the potential underlying factors. A systematic search strategy was employed on PubMed, Embase, MedRxiv, BioRxiv, and WHO COVID-19 databases from December 2019 till 15 September 2020. Two review authors independently assessed studies for inclusion, assessed quality, and extracted the data (PROSPERO registration #CRD42020182363). A total of 17 observational studies were included. The findings suggest that there was a reduction in the vaccination coverage and decline in total number of vaccines administered, which led to children missing out on their vaccine doses. An approximately fourfold increase was also observed in polio cases in polio endemic countries. Factors contributing to low vaccine coverage included fear of being exposed to the virus at health care facilities, restriction on city-wide movements, shortage of workers, and diversion of resources from child health to address the pandemic. As the world re-strategizes for the post-2020 era, we should not let a crisis go to waste as they provide an opportunity to establish guidelines and allocate resources for future instances. High-quality supplementary immunization activities and catch-up programs need to be established to address gaps during the pandemic era.

## 1. Introduction

The emergence of the novel severe acute respiratory syndrome coronavirus 2 (SARS-CoV-2) led the world into a crisis of unprecedented scale and scope [1]. As of 9 December 2020, more than 68 million confirmed cases and 1566 thousand deaths have been reported across 219 countries and territories, causing morbidity, mortality and societal disruption on a global scale [2]. As governments across the world attempted to regulate the outbreak by implementing population-wide lockdowns, closing borders and halting mass gatherings, experts started worrying about the indirect health impacts. These disruptions are very likely to threaten the progress of various programs, including immunization campaigns, which have proven to be a valuable and cost-effective public health intervention to date [3]. In an attempt to mitigate the devastating impact of the COVID-19 pandemic, the World Health Organization (WHO) issued guidelines calling for a temporary suspension of the operations of mass immunization programs across the globe [4,5]. While global vaccination programs were sliding backward even before this era [6], the related disruption is occurring at a scale larger than any since the advent of immunization programs in the 1970s. Consequences of such a falling through can be extensive, and could possibly even impact the future delivery of a COVID-19 vaccine.

According to the data collected by the WHO, the United Nations Children’s Fund (UNICEF), the Global Alliance for Vaccines and Immunizations (GAVI), and the Sabin Vaccine Institute, the suspension of vaccination services in over 68 countries have put at least 80 million children under the age of one at risk [7,8]. In low- and middle-income countries (LMIC) with health systems already under strain, even temporary disruptions can leave a trail of devastating health, opening the door to a possible resurgence of other illnesses [9]. An analysis by GAVI estimated that a further 24 million individuals who were hitherto protected through vaccinations are now at risk, as approximately 90 mass vaccination campaigns had been postponed [7,8,9,10]. With a continued decline in vaccine coverage rates, cases of intensified outbreaks of measles, diphtheria, pertussis, and other vaccine preventable diseases (VPDs), are drawing headlines [11], and multiple cases of polio and diphtheria have been reported across Pakistan and Afghanistan [12]. Measles is rising around the globe [13,14,15], dengue is flaring in regions of Latin America [16] and the Amazonian region [17], and countries in Africa find themselves under immense strain as they concurrently contend with outbreaks of measles and Ebola [18]. These outbreaks are a stark reminder that even during a pandemic, public health challenges are just as important, if not more. Drawing on historical and epidemiological analyses, and data from recent modelling exercises, has underscored the importance of maintaining essential health services including immunization [19,20]. The WHO predicted an increase in mortality caused by malaria in endemic regions [21]. Similarly, historical evidence from the past global disease outbreaks (e.g., diphtheria in the former Soviet Union in 1990–1996), wars and infectious threats have shown that lapses in primary care access and shifting attention away from routine health services lead to an increase in morbidity and mortality [22]. As COVID-19 triggered a similar breakdown of immunization systems, future of a hard-fought struggle to prevent mortality from VPDs is at stake.

In this systematic review, we illustrate the potential impact of the COVID-19 pandemic on immunization coverage across the globe and identify the factors contributing to the disruption. The findings from this study would provide insights for the impact of pandemics on existing vaccination strategies and hence help inform policy for effective planning for future pandemics and other crises.

## 2. Methods

This systematic review aimed to assess the magnitude of the impact of the COVID-19 pandemic on existing vaccination programs, and identify the factors associated with the disruption. We adhered to the guidelines established by Preferred Reporting Items for Systematic Reviews and Meta-Analyses (PRISMA; Appendix A) [23], and the title has been registered with PROSPERO (International Prospective Register of Systematic Reviews) under the identifier # CRD42020182363.

A search strategy was devised using appropriate, medical subject heading (MeSH), and free text terms to identify published (including pre-print) experimental, observational, and qualitative studies that have evaluated the impact of the COVID-19 pandemic on routine and/or mass immunization programs and campaigns. We also included studies exploring the factors associated with disruption in immunization programs and campaigns. Our search was designed to include all populations from both high-income countries and low- and middle-income countries impacted by COVID-19. We excluded opinion pieces, systematic reviews, commentaries, and news articles. Epidemiological studies that only identified the disruption of immunization coverage without reporting the impact on coverage and other outcomes of interest were also excluded. Studies pertaining to other coronavirus-related illnesses, such as Middle East respiratory syndrome (MERS) or SARS CoV-1, were excluded. Multiple reports from the same region were combined in an attempt to avoid duplicate data. For our review, we used the following definitions for routine and mass immunization. Routine immunization (RI) is the process of vaccinating in a sustainable, reliable manner, before exposure to the disease and using all vaccines in the national vaccination schedule to ensure every person is fully immunized against VPDs [24]. Mass immunization or supplementary immunization activities (SIAs) complement routine coverage and act as a catalyst to boost immunization coverage in a defined geographical area and/or area having low coverage. It delivers immunization to a large group at one or more locations over a short interval of time and is employed during National Immunization Days conducted in LMICs as part of the Expanded Program on Immunization (EPI) [25].

The following principal sources of electronic reference libraries were searched to access the available data: PubMed, EMBASE, the WHO COVID-19 database, and Google Scholar. The websites MedRxiv (https://www.medrxiv.org), and BioRxiv (https://www.biorxiv.org) were also searched for pre-print papers. We also searched the websites of selected development agencies or organizations such as GAVI, the vaccine alliance, UNICEF, and the WHO. No language restrictions were applied and the review included all studies published since December 2019. The last search date was 15 September 2020, and the search strategy is reported in Appendix A. The cited references of retrieved articles and annotated bibliographies were cross-checked for more eligible studies. All studies identified were uploaded into the Covidence-Systematic Review Software for screening [26] and then independently sifted for relevance by two review authors (RN and ZSL), and discrepancies were resolved by discussion. Multiple reports of the same study were collated to ensure each study was the unit of interest in this review. Data were independently extracted from eligible studies by two authors (RN and FS) using the pre-designed extraction sheet. Information was extracted on year of publication, study design, geographical setting, sample size, type of immunization campaign (routine or mass) and its coverage, i.e., pre-COVID-19 and during COVID-19 (where reported), and factors impacting the campaigns. We planned to perform a meta-analysis, however due to the heterogeneity in the outcome reported it was not possible, and we have descriptively summarized the findings from the included studies. We have synthesized the findings from the studies included under the following outcomes:Impact on routine immunization (including immunization coverage; and pediatric clinic visits);Impact on mass immunizations or supplementary immunization activities; andFactors affecting immunization coverage

Methodological quality of included studies was assessed using the criteria specified by the National Heart, Lung, and Blood Institute [27].

## 3. Results

A total of 3588 studies were identified from database searches. After excluding duplicates and articles that did not meet the inclusion criteria, we obtained 130 articles with full-texts for further evaluation, where another 113 were excluded as ineligible. We included a total of 17 observational studies [28,29,30,31,32,33,34,35,36,37,38,39,40,41,42,43,44] (Figure 1).

### 3.1. Characteristics of the Included Studies

Of the 17 observational studies, 10 explored immunization campaigns in high income countries (HICs) [29,30,32,36,37,38,39,40,42,44] and the remaining seven in LMICs [28,31,33,34,35,41,43]. Of the studies from HICs, five were conducted in the United States [30,32,36,39,42], two in the United Kingdom [37,40], and one each in France [44], Italy [29], and Spain [38]. Of those from LMICs, four were from Pakistan [28,33,35,43], one study was conducted across two countries namely, Afghanistan and Pakistan [34], and one each was from South Africa [41] and Sierra Leone [31] (Table 1).

As indicated in Table 2, fifteen studies focused on RI campaigns and two on SIAs. Two studies reported on polio cases [28,34]; four studies assessed the coverage of Measeles, Mumps, and Rubella (MMR) vaccines during the pandemic period compared to pre-pandemic times [32,33,37,44]; while the remaining studies explored the impact on vaccination coverage as a whole. Four studies compared the number of vaccine doses administered in 2020 to the preceding years [30,31,38,43]; two studies employed the number of vaccines ordered by physicians as proxy measures to assess vaccination coverage [32,39]; four studies highlighted the decline in pediatric hospital and clinical activity during the COVID-19 period [29,35,40,41]; while three studies assessed the willingness of health care providers to provide vaccination services during the pandemic [29,31,42].

Of all of the studies included, six studies used state or national level data [32,33,35,36,40,42]. Data sources included electronic health records, reports by national vaccine programs and surveillance systems, and state registries. Eight studies reported relevant project data [29,30,31,34,38,41,43,44], while others did not specify the source of data [28,38,40]. The quality of the studies included using the outlined criteria is reported in Table 3. On methodological quality, studies performed well in describing the research context, however, for other quality domains, either the information was missing or not reported at all.

### 3.2. Impact on Routine Immunization 

#### Immunization Coverage

A total of 12 studies analyzed the impact on immunization programs during COVID-19 [29,30,31,32,33,36,37,38,39,42,43,44]. Studies assessed immunization coverage during COVID-19 compared to the preceding years. The EPI-PHARE report from France contrasted the number of drugs delivered during the COVID-19 period (from 16 March to 10 May 2020) with the same weeks in 2018 and 2019. Over the course of the pandemic, vaccines delivered for infants fell by almost a quarter (−28.9%) and deliveries of MMR and human papilloma virus (HPV) vaccines dropped more than half (−50.8% and −78.1%, respectively) [44]. The Spanish Association of Pediatrics, similarly, recounted a decline in vaccination coverage ranging from 5% to 60%, in March 2020 compared to the monthly average for the period of January 2019 to February 2020 [38].

In England, the observed change in vaccinations counts in 2020 compared to 2019 varied over the course of the COVID-19 outbreak [37]. A decline in doses administered during the first 17 weeks of 2020 compared to 2019 was observed for both MMR (3.7% lower, 95% Cl (−3.8 to −3.6)) and hexavalent (3.5% lower, 95% Cl (−3.7 to −3.4)) vaccines. The effect was largely felt the same weeks strict social distancing measures were implemented (Week 13, 20 March 2020); following which MMR doses sharply decreased to a low point of −24.2% from 14.7% in week 12). Interestingly, the decline in doses of hexavalent vaccines did not accentuate with introduction of social distancing measures (week 13: −5.1%).

Studies conducted in the USA outlined a substantial impact on immunization across the country (Table 2). Chanchalani et al. [32] and Santoli et al. [39] highlighted a notable decline (approximately 75%) in vaccines ordered by doctors since January when the first person with COVID-19 was identified. Data analyzed for this subsection was provided by the vaccines for children (VFC) program provider order data. The steep decline in vaccination rates was notable for both non-influenza childhood (42%) and measles containing vaccine (50%). Consistent with reports of decreased vaccine counts across the country, data from state registries of Michigan and New York observed declines in vaccine doses to a similar degree; Michigan reported an approximately 18.5% drop during January–April 2020 [30] while the New York registry reported an almost 50% drop during COVID-19 (1 March to 27 June 2020), compared to the pre-COVID-19 era (years 2018–2019) [36]. Comparable to other countries, the largest decrease in vaccine coverage corresponded to the increase in severity of COVID-19 and the growing strictness of the lockdown, with rates of vaccination coverage improving as the course of the pandemic lightened, this impact was not just limited to HICs. Similar impact was observed in a rural area of Sierra Leone [31]. A retrospective analysis of a health facility during 1 March–26 April 2020 compared to the same period last year (1 March–26 April 2019) reported that, despite the COVID-19 pandemic, the health care center continued to function and offer their services, but the center reported a decline in children visiting the clinic for vaccinations of 57%. In an observational study based in Karachi, Pakistan, Chandir et al. underscored the inequity in vaccination coverage [33]. Data of a registry employed to track vaccination status was analyzed during baseline (4 Feb 2020, to 22 Mar 2020) and compared to the COVID-19 lockdown (23 Mar 2020 to 9 May 2020) and post COVID-19 lockdown (10 May 2020 to 6 Jun 2020) period. A geographic analysis illustrated considerable spatial heterogeneity. Slum and densely-packed squatter settlements—high-risk regions with historically low immunization coverage—were the worst affected. GAVI, the vaccine alliance, in their fortnightly situation report presented a similar picture across Pakistan [43]. With outreach sessions suspended, coverage dropped by more than 50% for all vaccines. Campaigns were further impacted after several polio workers contracted COVID-19 and as an inadvertent consequence of lockdown and flight disruptions, large parts of the country also reported running out of routine vaccines.

While some studies reported age distribution for patients, no specific pattern or trend was apparent. Bramer et al. observed a decline in all milestone age cohorts, except for birth-dose hepatitis B coverage which is customarily administered in hospitals [30]. The five-month age cohort observed a decline in vaccination status from approximately two-thirds (67.3%) of children during 2016 to 2019 to fewer than half (49.7%) in May 2020. In Karachi, the six-week old age group suffered the greatest drop (60%) in immunization coverage, a decline that could eventually lead to an increase in number of zero-dose children [33]. In contrast, Santoli et al. observed a less prominent decline in measles containing vaccines among children aged ≤24 months than among older children [39], and another study reported practices prioritizing offering vaccination to infants and toddlers [42]. This could suggest that system-level strategies to prioritize child care and immunization for the younger age group are robust.

Visits to clinic: Five studies assessed the disruption of pediatric clinical and hospital activities [29,33,35,40,41]. A pattern of overall decline was observed (Table 2). In South Africa, health visits declined by over 50% immediately after the lockdown (11.8 to 4.5 visits/day/clinic) was implemented, but partially bounced back post-lockdown (+1.1 visit/clinic/day) [41]. Likewise, Kirmani et al. recounted a sharp drop (1800 to 400) in the number of clinic visits the same week a strict lockdown was announced [35]. Parents reported fear of possible COVID-19 exposure to their children at health care facilities ultimately leading to the decline, and almost 60% of the parents had cancelled their child’s appointment [40]. In another study, almost all (98.2%) of the pediatricians surveyed reported a general decline in outpatient visits. Additionally, one study reported a decline (53%) in the mean number of daily immunization clinic visits during the lockdown compared to 6 months before COVID-19 [33]. A survey of health care practices underscored their willingness to operate and provide services during the lockdown [42]. Likewise, a study of Tuscan pediatricians reported over 90% doctors continued to vaccinate during the period [29].

### 3.3. Impact on Mass Immunization or Supplementary Immunization Activities (SIAs)

Two included studies highlighted the breakdown of polio campaigns [28,34] (Table 2) since late March 2020, a standstill in immunization was observed amid the COVID-19 outbreak. This outbreak has coincided with growing cases of polio in Afghanistan and Pakistan. As of June 19, 2020, 54 polio cases were reported this year—an approximate fourfold increase from 12 cases in 2019 [34]. Of these, 78% were detected in Pakistan and 22% in Afghanistan. A closer inspection reveals that polio cases were already running high—40 cases in Jan 2020 vs. 9 in Jan 2019—even before Global Polio Eradication Initiative (GPEI) suspended SIAs in March 2020 [34].

### 3.4. Factors Affecting Immunization Coverage

Nine studies reported factors affecting immunization coverage. The disruptions to immunization services were severe and were due to an interplay of multiple factors. Even when services were offered, people were concerned about being exposed to COVID-19 at health care facilities, fearing contacting the virus in waiting rooms and attempting to avoid contact with individuals with COVID-19 [29,31,38]. Additionally, they were reluctant to leave their homes and faced transport interruptions and restrictions on movements secondary to city-wide lockdowns [32]. Concerns of overburdening the already overstretched health care system also played a role [40]. In some regions, health workers were also unavailable because of restrictions on travel, redeployment to COVID-19 response activities, as well as the lack of protective equipment [33]. Furthermore, shuttering of educational facilities translated into a serious disruption in RI services that were previously delivered in school settings [28,37,38,45]. Thus, it appears that the COVID-19 pandemic and resultant chaos created a climate of fear and contributed to the unprecedented scale of disruptions of delivery and uptake of immunization services.

## 4. Discussion

Findings from this review show that the total number of vaccinations administered across all study regions began to decline precipitously during the COVID-19 pandemic. This drop in vaccinations administered coincides with both the WHO declaration describing the COVID-19 outbreak identified in Wuhan, China as a pandemic on March 11, 2020, and implementation of full national lockdowns, i.e., stay-at-home orders for regions or entire countries. As hospitals brace for an onslaught of critically ill patients and routine services worldwide are disrupted, health experts have had to shelve immunization campaigns for COVID-19. 

COVID-19 has affected virtually all countries in the world, regardless of their economic power, but represents a particular threat for LMIC’s for battling the pandemic alongside pre-existing challenges, including efforts to control VPDs and the capacity of health systems to manage the additional burden of COVID-19. Whilst the risk of COVID-19-associated mortality for children has so far been low and trivial in both LMICs and HICs [46,47], the children in LMICs are more likely to be disproportionately affected for several reasons [48]. Health systems in LMICs are under-resourced and weaker, and risk factors for poor health outcomes, such as malnutrition, poor sanitation, nutritional anemia, or human immunodeficiency virus (HIV) exposure or infection, are overwhelmingly more prevalent in LMICs than HICs [49,50]. Moreover, more than 90% of schools across the world faced closure, and education was interrupted due to the pandemic [51]. This further worsened inequalities in seeking education as many children in LMICs lack resources to adopt alternate arrangements [52]. With the risk of contracting communicable disease heavily skewed towards LMICs [53] and existing challenges compounded by the COVID-19 crisis, the long-term impact as a result of vaccine disruption would be severe in LMICs. Abbas et al. [54] weighed out the risks and benefits of reopening immunization clinics in the midst of the COVID-19 pandemic for LMICs in Africa. For every death attributable to SARS-CoV-2 infections acquired during RI clinic visits, 84 deaths from VPDs could be prevented. Hereby, outweighing the excess risk of COVID-19 deaths associated with vaccination clinic visits. The analyses by Zar et al. [9] provided further insight into the global health inequity. Reiterating that solutions for COVID-19, especially among the global poor, cannot include forgoing vaccinations. But this caveat is not only for lower-income countries. Despite the overall high rate of coverage of essential child immunization, HICs are also insecure [55], as we attempt to highlight in our review. This high coverage grants protection for some VPDs attenuating the immediate effects of vaccine disruption, however even short gaps in vaccination coverage can result in outbreaks of VPDs. Postponement of campaigns comes at a cost. Vaccinations are time sensitive, and if children are not immunized within the correct age-window, they forego benefits of life long immunity [56]. Whole cohorts of children may be left unprotected.

Studies also demonstrated regional variation in the decline of vaccination coverage [33,37]. This variation could be due to regional cultural differences affecting levels of compliance with health guidelines, level of concern regarding infection, and differences in socioeconomic status. The schism in immunization coverage between different wealth quintiles and urban/rural residence has been previously established in literature [57,58]. Stakeholders must not forget the disparities in vaccination coverage and remember that much remains to be done if the benefit of childhood immunization is to be maximized.

COVID-19 is a magnifying glass that has highlighted the larger threat of two existing public health challenges, namely polio and measles. The long-term trajectory of the pandemic is uncertain, and the risk of resurgence of epidemic-prone diseases is very high. Additionally, at the same time the health system’s capacity to perform a meaningful epidemiological analysis of the situation may also be impacted. Pakistan and Afghanistan were already struggling to eradicate polio [59], and with suspension of activities the risk of an epidemic is at the brink. In Pakistan, approximately 25,000 polio workers were diverted to help with the COVID-19 response [33]. Concerns of re-emergence of polio in previously polio-free countries escalated when Niger reported a new polio outbreak in April 2020 [60]. Discontinuation of polio vaccination programs may provide the virus a milieu to spread further and faster and could also result in worldwide export of infections. In addition, measles immunization campaigns have been suspended in 23 countries affecting almost 80 million eligible children [14]. Because of the highly infectious nature of measles, even a small decrease in routine measles vaccinations—a plausible result of lockdowns and disruption of health services—could lead to large and explosive outbreaks, which could raise the burden of childhood deaths significantly. It is imperative to ensure continuity of timely measles and polio vaccine administration for reaching the most vulnerable children and protecting them from common infectious disease. Policy makers must address the terrible calculus in determining how to proceed and capture the considerable loss of health from a wide range of diseases.

COVID-19 still presents as an evolving scenario with varying manifestations and emerging prevention and treatment strategies [61,62]. With this ever-evolving scenario, the pressure to meet spiraling health system needs is strong and a growing concern among virologists and public health experts, some wonder if the world has done more collateral damage in an attempt to curb the spread. Thus, while the drivers of low immunization coverage might lie outside of the health system, there are interventions that health policy makers could consider to mitigate disruptions and prioritize vaccination programs as avoiding vaccine-preventable deaths is critical for nations to preserve their gains in child survival.

The review has identified the following limitations. First, we identified a very limited number of studies on the impact of the COVID-19 pandemic on immunization coverage, particularly those that have reported the barriers or factors affecting the disruptions from users or providers perspective. Secondly, studies reported outcomes differently and therefore we could not pool or meta-analyze them. Third, the majority, but not all, compared the disruption of pandemic on vaccination coverage with pre-COVID vaccination coverage, and therefore we couldn’t report the actual impact in numbers or proportions. However, as the pandemic continues, there are limited data and not all studies reported pre-pandemic numbers for comparison. As we attempt to highlight the impact of COVID-19 on pre-existing immunization programs, factors such as the cost of vaccines and policies determining immunization have not been explored.

## 5. Conclusions

The COVID-19 pandemic has disrupted health and health systems worldwide, and most countries have still not recovered from the immediate effects of the increased mortality and morbidity due to SARS-CoV-2 infection. The global pandemic may affect the immunization program by leading to a decline in financial, human, and other resources, social-instability, limiting political attention and the country’s health priorities. This, in addition to the devastating consequences of the prolonged lockdowns, will challenge both HICs and LMICs for years to come, irrespective of their health infrastructure. Additionally, declines in vaccination coverage and refusal to vaccinate during the pandemic may adversely impact the dissemination of an imminent SARS-CoV-2 vaccine. It is essential to urgently address the basic needs of health care facilities and resume routine health services with appropriate COVID-19 preventive measures in place. The need to define an integrated strategy that could involve all entities to safeguard individuals and mitigate the impact of the outbreak is crucial.

## Figures and Tables

**Figure 1 ijerph-18-00988-f001:**
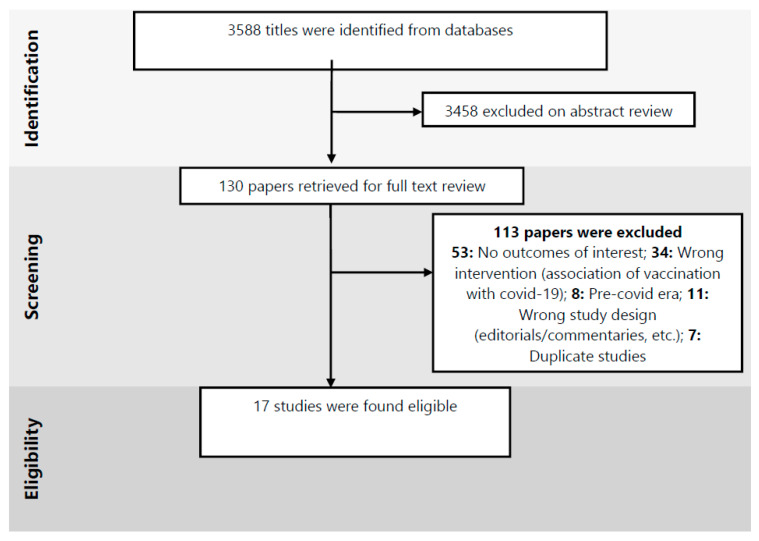
Preferred Reporting Items for Systematic Reviews and Meta-Analyses (PRISMA) flow diagram of study selection process.

**Table 1 ijerph-18-00988-t001:** Characteristics of studies included.

Author	Objective	Country/Region	Method of Retrieval	Source of Data Retrieval
**High income countries**				
Bechini 2020 [29]	Evaluate the impact of the COVID-19 epidemic on pediatric vaccinations administered	Tuscany, Italy, Europe and Central Asia	Members of Italian Federation of Pediatricians participated in a semi-structured online survey	Project-level data
Bramer 2020 [30]	i. Evaluate whether vaccination coverage has changed during the pandemic. ii. Changes in vaccine doses administered to children and the effects of those changes	Michigan, United States, North America	Michigan Care Improvement Registry	State-level: Data derived from a state-level immunization information system, i.e., Michigan Care Improvement Registry.
Chanchlani 2020 [32]	Discussed how limited access to health care, parental fear of seeking health care, school closures & financial instability as a result of COVID-19 may lead to increased morbidity and mortality.	United States, North America	Reports by US Vaccines for Children (VFC ) Program	National level—VFC Program is a federally funded program through the CDC”
Langdon-Embry 2020 [36]	Weekly number of routine vaccine administered to persons aged <24 months & 2–18 years in 2020 vs. 2019	New York City, United States, North America	Citywide immunization registry—2.7 million patients records	State-level: Data derived from a state-level immunization information system
McDonald 2020 [37]	Assess the early impact of COVID-19 on routine childhood vaccination in England to 26 April 2020.	England & Scotland, United Kingdom; Europe and Central Asia	EHR (electronic health records) records—electronic patient records for over 2600 primary care practices in the UK and over 35 child health providers	State/national level Data derived from a SystmOne software system.
Moraga-Llop 2020 [38]	Discusses aspects of recovery of vaccination coverage	Spain, Europe and Central Asia	Not defined	-
Santoli 2020 [39]	Assess the impact of the pandemic on pediatric vaccination in the United States.	United States, North America	VFC Program provider order data from CDC’s Vaccine tracking System.	National level—“The VFC Program and Vaccine Tracking System
Saxena 2020 [40]	Assess the impact of COVID-19 on routine immunizations	England & Scotland, United Kingdom; Europe and Central Asia	Not defined	Not defined
Vogt 2020 [42]	Assess the capacity of pediatric health care practices to provide immunization services to children during the covid-19 pandemic.	United States, North America	Survey of practices participating in the Vaccines for Children (VFC) program	Project data
Alain Weill 2020 [44]	Report quantifies the evolution of drug use prescribed in France, since the start of the epidemic	France; Europe and Central Asia	Number of users measured & compared to the ‘expected’ number of consumers-estimated from consumption data during the same weeks in 2018 and 2019	Project data; data from National Hospital Discharge Survey (NHDS)-EPI-PHARE reported dispensing of reimbursed drugs in pharmacies since the start of the epidemic
**Low- and middle-income countries**				
Haqqi 2020 [28]	Assess the impact on global polio eradication initiative	Pakistan, South Asia	Not defined	Not defined
Buonsenso 2020 [31]	Understand the potential impact of COVID-19 on child vaccinations in a typical poor peripheral area of Sierra Leone	Kent, Rural Western Area, Sierra Leone; Sub Saharan Africa	Cases retrospectively collected by the center’s health workers registry book	Project- level data
Chandir/GAVI 2020 [33]	Understand the impact of COVID-19 restrictions on routine immunization coverage in Karachi	Karachi, Pakistan, South Asia;	Used data from the provincial Electronic Immunization Registry.	Project- level data
Chard 2020 [34]	Summarizes progress toward polio eradication during 1 January 2018 and 31 March 2020	Afghanistan & Pakistan; South Asia	Global Polio Eradication Initiative (GPEI)	Project level—derived from GPEI
Kirmani 2020 [35]	Study the disruption in clinical activity at the Children’s Hospital	Karachi, Pakistan, South Asia;	Systematically studied the disruption in clinical activity	Project-level data
Siedner 2020 [41]	Assess the impact of the lockdown order in response to the COVID-19 epidemic in South Africa	Kwa-Zulu Natal, South Africa, Sub-Saharan Africa	Data collected by the Africa Health Research Institute (AHRI)	National-level data—data on clinic visitation at 11 public health clinics
GAVI 2020 [43]	Situation report #14	Pakistan, South Asia	Expanded program on immunization	Project- level data

Abbreviations: CDC: Centers for Disease Control, and Prevention; VFC: vaccine for children.

**Table 2 ijerph-18-00988-t002:** Outcomes reported in the included studies.

Author	Time Period	Vaccine	Outcome	
**Mass immunization or supplementary immunization activities (SIAs)**				
New polio cases				
Haqqi 2020 [28]	COVID-19 until June 2020	Polio	New polio cases	Polio cases as of June 2020: 51 cases
Chard 2020 [34]	1 January 2018–31 March 2020	Polio	New polio cases	Jan–March 2020: 54 cases- 12 in Afghanistan and 42 in Pakistan Jan–March 2019: 12 cases- 6 in Afghanistan and 6 in Pakistan
Routine Immunization				
Immunization coverage				
Bramer 2020 [30]	a point in time in May 2020 compared to points in time in May 2016–May 2019	HepB; rotavirus (Rota); Diphtheria, Tetanus, Pertussis (DTaP); Haemophilus influenzae type b; pneumococcal conjugate; inactivated poliovirus; MMR; varicella; hepatitis A	Doses of vaccines administered Percentage of children vaccinated	Number of doses administered Jan–April 2020 vs. 2018 & 2019: children aged ≤18 years decreased 21.5% children aged ≤24 months decreased 15.5% Percentage of infants and children vaccinated: 1 month: 2016–19: 82; 83; 82; 84, 2020: 85 3 months: 2016–19: 74; 75; 74; 65, 2020: 61 5 months: 2016–19: 66.6; 67.4; 67.3; 67.9, 2020: 49.7 7 months: 2016–19: 52; 54; 55; 54, 2020: 44 16 months: 2016–19: 52; 54; 54; 55; 2020: 40 19 months: 2016–19: 60; 62; 60; 2020: 5524 months: 2016–19: 45; 47; 49. 50, 2020: 43
Buosenso 2020 [31]	1 March 2019 to 26 April 2019 v1 March 2020 to 26 April 2020	Bacillus Calmette–Guérin (BCG), Oral poliovirus vaccines (OPV), Pneumococcal vaccine (PCV),Tetanus txoid (TT), pentavalent vaccine (PENTA): diphtheria, pertussis, tetanus, hep b, hemophilus, TT, Intermittent preventive treatment in infant	Vaccines administered at a health centre	BCG: Pre-covid 36; COVID: 17; Change −52.7; *p*-value < 0.0005 OPV0 Pre-covid 36; COVID: 17; Change −52.7; *p*-value < 0.0005 OPV1 Pre-covid 58; COVID: 17; Change −70.7; *p*-value < 0.0005 PENTA1 Pre-covid: 58; COVID: 17; Change −70.7; *p*-value: < 0.0005 PCV1 Pre-covid 58; COVID: 17; Change −70.7; *p*-value: < 0.0005 ROTA1 Pre-covid: 58; COVID: 17; Change −70.7; *p*-value: < 0.0005 OPV2 Pre-covid: 71; COVID: 15; Change −78.9; *p*-value: < 0.0005 PENTA2 Pre-covid: 71; COVID: 15; Change −78.9; *p*-value: < 0.0005 PCV2 Pre-covid: 71; COVID: 15; Change −78.9; *p*-value: < 0.0005 ROTA2 Pre-covid: 71; COVID: 15; Change −78.9; *p*-value: < 0.0005 IPTI1 Pre-covid: 49; COVID: 15; Change −69.4; *p*-value: < 0.0005 OPV3 Pre-covid: 67; COVID: 15; Change −77.6; *p*-value: < 0.0005 PENTA3 Pre-covid: 67; COVID: 15; Change −77.6; *p*-value: < 0.0005 PCV3 Pre-covid: 67; COVID: 15; Change −77.6; *p*-value: < 0.0005 IPTI2 Pre-covid: 44; COVID: 15; Change −65.9; *p*-value: < 0.0005 IPV Pre-covid: 67; COVID: 15; Change −77.6; *p*-value: < 0.0005 IPTI3 Pre-covid: 45; COVID: 22; Change −51.1; *p*-value: < 0.0005 Measles Pre-covid: 64; COVID: 22; Change −65.6; *p*-value: < 0.0005 Yellow fever Pre-covid 64; COVID 22; Change −65.6; *p*-value < 0.0005 Measles 2nd Pre-covid 49; COVID 8; Change −83.7; *p*-value < 0.0005
Chandir/GAVI 2020 [33]	6 months before the lockdown (23 Sept 2019–22 March 2020) vs. first 6 weeks of the lockdown (March 23–May 9, 2020)	General	Immunization coverage Visit to immunization clinic	Daily average number of immunization 6 months before COVID-19 lockdown: Total: 16, 649 Enrollments: 32% Follow-ups: 68% During COVID-19 lockdown: Total: 5836 Enrollments: 31% Follow-ups: 69% Immunization of children (aged 0–23 months): Baseline: 608,832 children Lockdown: 92,492 children decrease in mean number of daily immunization visits vs baseline: Lockdown (March–May 9): −52.8% (5184 to 2450) Post-lockdown (May 10–June 2020): −27.2% (5184 to 3772). 2734 children per day missed routine immunization during the lockdown Immunization centre: 16% of 321 immunization centres had no flow reduction in immunization doses: 88.6% for outreach; 38.7% for fixed Vaccinators attending work: Baseline: 91.6% Lockdown: 78.7%
Langdon-Embry 2020 [36]	2020 vs. 2019	General	Doses administerednumber of facilities that administered at least one vaccine in 2020	Mar 1: First confirmed COVID-19 case in NYC <24 months: 2019: 32,000; 2020: 32,0002–18 years: 2019: 19,000; 2020: 20,000 Mar 13: National emergency declared <24 months: 2019: 35,000; 2020: 31,0002–18 years: 2019: 20,000; 2020: 16,000 Mar 22: NYC on PAUSE—required New Yorkers to stay at home <24 months: 2019: 33,000; 2020: 15,0002–18 years: 2019: 21,000; 2020: 3000 April 5–11: <24 months: 62% decrease: 2019: 33,000; 2020: 13,0002–18 years: 96% decrease: 2019: 24,000; 2020: 1000 facilities administering at least one vaccine, <24 months: 46% decrease, from 900 in 2019 to 488 in 2020 facilities administering at least one vaccine, 2–18 years: decreased 78%, from 1238 in 2019 to 275 in 2020 April 19–25: number of new COVID-19 cases declined <24 months: 2019: 27,000; 2020: 20,0002–18 years: 2019: 25,000; 2020: 3000 June 21–27: facilities administering at least one vaccine, <24 months:increased 69% from the lowest point to 825 Vaccines administered to persons 2–18 years:35% fewer vaccines (17,947 doses versus 27,405).
McDonald 2020 [37]	first 17 weeks of 2019 and 2020:	i. Hexavalent: DTaP, polio, Hib and Hep B ii. MMR	Immunization coverage	1. Hexavalent: i. 2019: 69,568; ii. 2020: 67,116, Percentage change: −3.5 (−3.7 to −3.4) 2. MMR: i. 2019: 65, 341; ii. 2020: 61, 832. Percentage change: −3.7 (−3.8 to −3.6) Week 1–9: prior to social distancing Hexavalent: −5.8% (95% CI—6.0 to −5.5%) MMR: −1.0% (95% CI −1.1 to −0.9%) Week 10–12: instruction of self isolation by UK government Hexavalent: 4.4% (95% CI −4.8 to −4.0%) MMR: −7.2% (95% CI −7.7 to −6.7%) Week 13: Strict social distancing introduced Hexavalent: −5.1% (−5.9 to −4.4) MMR: −24.2 (−25.9 to −22.5) Week 13–15: Hexavalent: −6.7% (95% CI −7.1 to −6.2%) MMR: −19.8% (95% CI −20.7 to −18.9%)
Moraga-Llop 2020 [38]	COVID-19 era vs. Pre-COVID-19	Hexavalent, pneumococcus, meningococcus, MMR, Varicella, Men B, Rota, Tdap	Immunization coverage	Coverage decreased in all autonomous communities between 5% and 60%Reduction in vaccines administered (March 2020 vs. 2019) 2 & 4 mths- hexavalent, pneumococcus & meningococcus C: 8–13% 11-month booster doses -hexavalent and pneumococcus: 15% 12-months -MMR & meningococcus ACWY: 12% 15 months- varicella: 20% Rota virus: 19% Men B: 40% pertussis vaccination in pregnant women was not affected (−0.6%) decrease more evident for vaccines not funded: Valencian: 1st dose of Men B: −68.4% Andalusia: total doses of Men B: −39% Rotavirus: −18% decrease accentuated with age; Valencian Community: tetanus & diphtheria (>64 years of age): −67.5%. private pediatrics did not register the same effects
GAVI 2020 [43]	Mar–April 2019 vs. 2020	Pentavalent 3, Oral polio vaccine type 3 (OPV3), Measles, Bacillus Calmette–Guérin (BCG)	Immunization coverage	% target children covered: Pentavalent 2019: 89, 2020: 40 Oral polio vaccine (OPV) 2019: 88, 2020: 62 Measles 2019: 84, 2020: 61 BCG 2019: 88, 2020: 64
Alain Weil 2020 [44]	Lockdown: March 16–May 10, 2020Post-lockdown: May 11–17, 2020	Anti- human papilloma virus (HPV) Combined Penta/Hexavalent vaccines for infants: Hæmophilus influenzæ B, pertussis, poliomyelitis, tetanus, hep B MMR Anti-tetanus (excluding infants)	Drugs prescribed since the start of the epidemic	March 2–8: HPV: −3.6; Comb: −0.4; MMR: −13.6; Tetanus: +0.3 March 9–15: HPV: −6.8; Comb: −2.5; MMR: −12.3; Tetanus: −6.5March 16–22: HPV: −21.9; Comb: −3.6; MMR: −28.5; Tetanus: −26.2 March 23–29: HPV: −67.4; Comb: −23; MMR: −49; Tetanus: −64.9 March 30-April 5: HPV: −78.1; Comb: −28.9; MMR: −50.8; Tetanus: −77.3April 6–12: HPV: −74.4; Comb: −21.3; MMR: −50.8; Tetanus: −76.5 April 13–19: HPV: −69.6; Comb: −18.7; MMR: −46.9; Tetanus: −72.9 April 20–26: HPV: −45.6; Comb: + 6.1; MMR: −21.7; Tetanus: −55.2April 30–May 3: HPV: −45.6; Comb: + 6.1; MMR: −21.7; Tetanus: −55.2 May 4–10: HPV: −43.2; Comb: −5.6; MMR: −15.7; Tetanus: −48.4 Difference: HPV: −89,508; Comb: −44,171; MMR: −123,966; Tetanus: −446,580 Post-lockdown May 11–17: HPV: −34.4; Comb: −6.4; MMR: −4.7; Tetanus:−37,9
Vaccine orders				
Chanchlani 2020 [32]	6 Jan.–19 Apr. 2020, v. 2019	General, MMR, diptheria	decrease in orders	Decrease in vaccination: Apr. 5, 2020 vs Feb 16 (pre-coronavirus): 1. 50% for MMR 2. 42% for diphtheria and whooping cough
Santoli 2020 [39]	Period 1: 7 January–21 April 2019Period 2: 6 January–19 April 2020	DTaP; Hib; Hep A; Hep B; HPV: influenza; MMR; Meningococcal; Pneumococcal 13-& 23 valent; IPV; Rota; Tdap; Var	decrease in vaccine orders in period 2, compared to period 1	measles-containing vaccine administered: 21.5% decline (Mar, 2020)Children ≤24 months: March 9th (before US national emergency): 2000 March 16th (After US national emergency): 1100 Children >24 months through 18 years: March 9th (before US national emergency): 2250 March 16th (After US national emergency): 500 difference in doses of vaccines ordered by health care providers:non-influenza vaccines Jan 20th (first U.S. COVID-19 case reported): −500,000Apr 13th: −3,000,000 measles-containing vaccinesJan 20th: −100,000Apr 13th: −400,000
Health care providers providing immunization services during the pandemic				
Bechini 2020 [29]	Lockdown: 11 March 11–4 May 2020	Not identified	Activities & behavior of pediatricians Visits to clinic	1) 208 (93.3%) pediatricians continued to vaccinate 66 (31.7%): reduction in compliance to mandatory vaccines (hexavalent and MMRV) 88 (42.3%): reduction in compliance to non-mandatory vaccines 37 (17.8%) administrated only the first doses of the vaccines 82.2%) reported administering all the scheduled vaccine doses. 2) 15 (7%) had suspended vaccinations 98.2%- pediatricians reported a general decline in outpatient visits during the COVID-19 epidemic period
Vogt 2020 [42]	12–20 May 2020	Not identified	Practices providing immunization	Of 1933 practices: open: 1727 (89.8%) providing immunization services: 1397 (81.1%)
Visits to clinic				
Bechini 2020 [29]	Lockdown: 11 March–4 May 2020v pre-COVID era	Not identified	Visits to clinic	(98.2%) reported a general decline in outpatient pediatric visits during the COVID-19 epidemic period65.8% reported a more than 60% reduction in comparison with the situation before the COVID-19 pandemic.
Kirmani 2020 [35]	February 2020 to post-lockdown	Not identified	Visits to clinic	Week 7 (pre-COVID): 2600 Week 8 (26 Feb, First case of COVID reported): 2200Week 11 (21 March, Sindh Lockdown): 1800Week 13: 400Week 15: 1300Week 17: 1200Week 19: 1300
Saxena 2020 [40]	-	Not identified	Visits to clinic;Parental concern;Vaccine coverage	Children Visits to ED: fell by over 90% during April 2020 Parents expressed concerns about overburdening the NHS and fear of exposure to covid-19 when attending for vaccination. 60% of 752 health visitors surveyed in May 2020 reported families who had considered cancelling or postponing their child’s vaccinations
Siedner 2020 [41]	Pre lockdown: 60 days prior to lockdown Lockdown: 27th March–30th April’ 20: Post lockdown: 35 days after the lockdown.	Not identified	Visits to clinic	overall clinic visits during lockdown: −0.2, (95%CI −3.4, 3.1) visits/day/child-health clinic: pre-lockdown period: 11.8 (8.4, 15.1) lockdown period: 4.5 mean change: −7.2 (95%Cl −9.2, −5.3); p-value: <0.001 mean change post-lockdown: +1.1 (0.5, 1.7); p-value: 0.001Age-stratified: Total visits Jan–Apr. 2020: 6194 <1 year: 4270 (68.9%) 1−5 years:1786 (28.8%) 6–19 years: 103 (1.7%) 20–45 years: 29 (0.5%) >45 years: 6 (0.1%)children under 1: mean decrease of −5.3 visits, 95%CI −7.1, −3.61–5 years: mean decrease of −5.5 visits, 95%CI −6.8, −4.2

**Table 3 ijerph-18-00988-t003:** Quality of included studies.

Study ID	1. Study Question or Objective	2. Eligibility/ Selection Criteria	3. Representative Participants	4. Eligible Participants Met Entry Criteria	5. Sample Size	6. Test/ Service/ Intervention Clearly Described	7. Outcome Measures	8. Outcomes Blinded	9. Loss to Follow-up	10. Statistical Methods	11. Interrupted Time-Series Design	12. Use of Individual-Level Data
Haqqi 2020 [28]	Yes	No	No	NR	NR	NR	No	N/A	N/A	No	No	No
Bechini 2020 [29]	Yes	Yes	Yes	NR	CD	Yes	Yes	N/A	N/A	No	No	No
Bramer 2020 [30]	Yes	Yes	Yes	Yes	Yes	Yes	Yes	N/A	N/A	No	No	No
Buonsenso 2020 [31]	Yes	Yes	Yes	NR	CD	Yes	Yes	N/A	N/A	Yes	No	No
Chanchlani 2020 [32]	Yes	Yes	NR	N/A	NR	NR	No	N/A	N/A	No	No	No
Chandir/GAVI 2020 [33]	Yes	Yes	NR	NR	NR	Yes	Yes	N/A	N/A	No	No	No
Chard 2020 [34]	Yes	No	No	NR	Yes	Yes	CD	N/A	N/A	No	No	No
Kirmani 2020 [35]	Yes	Yes	NR	NR	NR	N/A	Yes	N/A	N/A	No	No	No
Langdon-Embry 2020 [36]	Yes	Yes	Yes	Yes	Yes	Yes	Yes	N/A	N/A	No	No	No
McDonald 2020 [37]	Yes	Yes	Yes	Yes	Yes	Yes	Yes	N/A	N/A	CD	NR	No
Moraga-Llop 2020 [38]	Yes	No	CD	N/A	NR	NA	NR	N/A	N/A	No	No	No
Santoli 2020 [39]	Yes	Yes	Yes	Yes	Yes	Yes	Yes	N/A	N/A	No	No	No
Saxena 2020 [40]	Yes	NR	NR	NR	NR	No	No	N/A	N/A	No	No	No
Siedner 2020 [41]	Yes	Yes	Yes	Yes	Yes	Yes	Yes	N/A	N/A	Yes	Yes	No
Vogt 2020 [42]	Yes	Yes	Yes	NR	Yes	Yes	Yes	N/A	N/A	No	No	No
GAVI 2020 [43]	Yes	No	NR	NR	CD	N/A	NR	N/A	N/A	No	No	No
Alain Weill 2020 [44]	Yes	Yes	Yes	N/A	NR	CD	NR	N/A	N/A	No	No	No
Detailed Criteria: Was the study question or objective clearly stated? Were eligibility/selection criteria for the study population prespecified and clearly described? Were the participants in the study representative of those who would be eligible for the test/service/intervention in the general or clinical population of interest? Were all eligible participants that met the prespecified entry criteria enrolled? Was the sample size sufficiently large to provide confidence in the findings? Was the test/service/intervention clearly described and delivered consistently across the study population? Were the outcome measures prespecified, clearly defined, valid, reliable, and assessed consistently across all study participants? Were the people assessing the outcomes blinded to the participants’ exposures/interventions? Was the loss to follow-up after baseline 20% or less? Were those lost to follow-up accounted for in the analysis? Did the statistical methods examine changes in outcome measures from before to after the intervention? Were statistical tests done that provided p values for the pre-to-post changes? Were outcome measures of interest taken multiple times before the intervention and multiple times after the intervention (i.e., did they use an interrupted time-series design)? If the intervention was conducted at a group level (e.g., a whole hospital, a community, etc.) did the statistical analysis take into account the use of individual-level data to determine effects at the group level?												

CD: cannot determine; NA: not applicable; NR: not reported.

## Data Availability

All data is available upon reasonable request.

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
