# Peer review of "The Impact of the COVID-19 Pandemic on Immunization Campaigns and Programs: A Systematic Review"

_ijerph, 2021, doi:10.3390/ijerph18030988_

Round 1

Reviewer 1 Report

In the manuscript presented for review, Dr. Lassi and colleagues provided an important systematic review that clearly showed that vaccination coverage during the COVID-19 pandemic is significantly reduced, with other infectious diseases becoming more common.

The article is written carefully, but requires a few changes:

1. the introduction is definitely too long and must be shortened,
2. the publication does not contain figures and tables, despite the fact that the authors cite them in the text of the article,
3. I can't find bias risk assessment in individual studies either. Regardless of the fact that the authors did not perform the mat-analysis, such an assessment must be included in the paper,
4. the search strategy should be supplemented with the number of results for at least one database,
5. Please quote the following articles: doi: 10.3390/pathogens9060493; doi: 10.3390/pathogens9060501).
6. Literature is extensive which is fine, please check the formatting.

Author Response

Reviewer 1

In the manuscript presented for review, Dr. Lassi and colleagues provided an important systematic review that clearly showed that vaccination coverage during the COVID-19 pandemic is significantly reduced, with other infectious diseases becoming more common.

The article is written carefully, but requires a few changes:

  1. the introduction is definitely too long and must be shortened,

Answer to reviewer’s comment: Thank you for your feedback. We have revisited the introduction section and have shortened it.

  1. the publication does not contain figures and tables, despite the fact that the authors cite them in the text of the article,

Answer to reviewer’s comment: Thank you for your feedback. Sorry that you haven’t had an access to those but we did provide those with our submission. We are now providing them again with this revised version from Page 14 to 21

  1. I can't find bias risk assessment in individual studies either. Regardless of the fact that the authors did not perform the mat-analysis, such an assessment must be included in the paper,

Answer to reviewer’s comment: Thank you for your feedback. Sorry that you haven’t had an access to the risk of bias assessment table. We are providing them again with this revised version as table 3 on page 21

  1. the search strategy should be supplemented with the number of results for at least one database.

Answer to reviewer’s comment: We have added the numbers to the Supplementary file 2

  1. Literature is extensive which is fine, please check the formatting.

Answer to reviewer’s comment: we have revisited the entire manuscript and formatted where required. 

Reviewer 2 Report

The review presents an important question but needs considerable work before meriting publication to the standards expected of IJERPH.

The methodology is not described in sufficient detail and needs considerable development. The exclusion criteria for studies needs to be properly described in a lot more detail.

Diagrams are essential here within the methods to help communicate what studies were lacking which justified exclusion. A standard PRISMA diagram would be advised.

A table of the papers which were able to be included with further detail on which disease/location/main findings/other classification criteria would be useful. This should be grouped by disease. 

The results need figures and tables summarising the information. The figures should be demonstrating a numerical output (i.e. decline in vaccination rate) which allows the reader to synthesise the information. This should be broken down into HMIC and LMIC where applicable or split into groups which reflect the discussion of the main findings.

At this current presentation, it is almost impossible to follow as there is a lot of regurgitation of individual paper's main findings with insufficient binding narrative which allows the reader to draw interesting insights or concrete take home messages.

Author Response

  1. The methodology is not described in sufficient detail and needs considerable development. The exclusion criteria for studies needs to be properly described in a lot more detail.

Answer to reviewer’s comment: Thank you for your feedback. To ensure detailed description, changes have been made. For reference: Line: 103-121

  1. Diagrams are essential here within the methods to help communicate what studies were lacking which justified exclusion. A standard PRISMA diagram would be advised.

Answer to reviewer’s comment: Thank you for your feedback. Sorry that you haven’t had an access to the figures and tables, we are providing them again and at the end of the manuscript with fig 1 as PRISMA on page 21  

  1. A table of the papers which were able to be included with further detail on which disease/location/main findings/other classification criteria would be useful. This should be grouped by disease. 

Answer to reviewer’s comment: The table with the included studies is provided as table 1 and now we are putting this at the end of the manuscript on page 15-21

  1. The results need figures and tables summarising the information. The figures should be demonstrating a numerical output (i.e. decline in vaccination rate) which allows the reader to synthesise the information. This should be broken down into HMIC and LMIC where applicable or split into groups which reflect the discussion of the main findings.

Answer to reviewer’s comment: Sorry that you did not get access to the figures/tables. The tables have these information and we are now providing them again at the end of the manuscript (page 14 to 21) and specifically see table 2 (page 24).

  1. At this current presentation, it is almost impossible to follow as there is a lot of regurgitation of individual paper's main findings with insufficient binding narrative which allows the reader to draw interesting insights or concrete take home messages.

Answer to reviewer’s comment: thank you for this comment, we have now revisited the manuscript and edited to give it a flow for readers.

Reviewer 3 Report

Zohra S Lassi et al proposed a systematic review of immunization programs in different countries.

Comment 1: Immunization programs are varied in different countries based on different policies. There is little evidence in this study to show the impact of COVID-19 or other factors that determine the differences in different countries.

Comment 2: The cost of vaccines is very important for parents to decide what vaccines do they children will be taken in many circumstances. If this study only report observational studies, it is not enough.

Comment 3: Figures and Tables are mentioned in the manuscript, but I could not find any in the manuscript and the supplementary.

Comment 4: And statistical methods/results that merge data from different studies are lacking.

Author Response

Reviewer 3

Zohra S Lassi et al proposed a systematic review of immunization programs in different countries.

  1. Comment 1: Immunization programs are varied in different countries based on different policies. There is little evidence in this study to show the impact of COVID-19 or other factors that determine the differences in different countries.

Answer to reviewer’s comment: In this study, we attempt to study the pre and during/post COVID-19 impact on immunization programs across the globe. We do not compare the difference in countries but merely attempt to highlight the impact of COVID-19 on preexisting programs and their policies. This would have been a very useful information, but with the current limited data – this would be hard to infer. For clarity we have added this under limitations on page 14, lines: 376 - 385

  1. Comment 2: The cost of vaccines is very important for parents to decide what vaccines do they children will be taken in many circumstances. If this study only report observational studies, it is not enough.

Answer to reviewer’s comment: The objective of this systematic review was to review the uptake and coverage that has been affected because of COVID pandemic by comparing the coverage before and during/post COVID times. This could have been an important factor but was not found in the studies included.

  1. Comment 3: Figures and Tables are mentioned in the manuscript, but I could not find any in the manuscript and the supplementary.

Answer to reviewer’s comment: Thank you for your feedback. Sorry that you haven’t had an access to those but we did provide those with our submission. We are providing them again with this revised version at the end of the manuscript from page 14 to 21  

  1. Comment 4: And statistical methods/results that merge data from different studies are lacking.

Answer to reviewer’s comment: Due to the heterogeneity in data, it was not possible to merge data and execute any statistical analysis in our review. This has been reported on page 3, lines 141-144.

Reviewer 4 Report

This manuscript highlights the important public health issue during COVID-19 pandemic. A lot of study focuses on COVID-19 and sometimes the impact of pandemic on other health care issues tend to be fallen behind. I think this manuscript is worth to be published, but it required extended revision. Please check the PRISMA checklist again.

Introduction

Please specify the objectives. (PICOS)

For example, you need to address your target population (e.g. LMIC?)

Please specify why you need to conduct this study.

What is your current knowledge about the impact of COVID-19 pandemic on vaccination program from the previous study?

Methods

Please specify inclusion and exclusion criteria.

In outcome measure, you include inequality in immunization coverage. How do you address the inequality?

Results

Overall

You write Figure and Tables in the text. But I could not find them in the manuscript.

In the results session, you only need to present the results from the included studies. You mixed too much information.

For example, in line 214-216, this is something you need to write in discussion part.

Other comments

Impact on routine immunization

You only write the rate of decline in vaccine coverage, but it is better to include the vaccination coverage before and during pandemic.

Line202-207

I cannot understand this part. You said that the impact was not just limited to HICs in first sentence, but you said the health care center observed no reductions in activities.

Inequality in vaccination coverage

This section actually does not address the inequality of the vaccine coverage.

Impact on mass immunization or SIAs

In this section, you just present the increase in polio cases. How did you know this is the effect of mass immunization or SIAs?

Author Response

Reviewer 4

This manuscript highlights the important public health issue during COVID-19 pandemic. A lot of study focuses on COVID-19 and sometimes the impact of pandemic on other health care issues tend to be fallen behind. I think this manuscript is worth to be published, but it required extended revision. Please check the PRISMA checklist again.

  1. Introduction

Please specify the objectives. (PICOS)

For example, you need to address your target population (e.g. LMIC?)

Answer to reviewer’s comment: Thank you for you comment. We have edited this under both introduction and methods: 86-96 and 107-114

  1. Please specify why you need to conduct this study.

Answer to reviewer’s comment: Thank you for your feedback. We have now added this at the end of the background (line 94-96) “The findings from this study would provide insights for the impact of pandemic on existing vaccination strategies and hence help inform policy for effective planning for future pandemics and other crisis.”

  1. What is your current knowledge about the impact of COVID-19 pandemic on vaccination program from the previous study?

Answer to reviewer’s comment: To the best of our knowledge, this specific topic has not been reviewed before. In our review, we have attempted to explore all studies that have reported the impact on vaccination programs.

Methods

  1. Please specify inclusion and exclusion criteria.

Answer to reviewer’s comment: Thank you for your feedback. Required changes can be found at line 107 to 114

  1. In outcome measure, you include inequality in immunization coverage. How do you address the inequality?

Answer to reviewer’s comment: In order to avoid any confusion, we have removed this as an outcome measure. Line 145-146, page 4

Results

Overall

  1. You write Figure and Tables in the text. But I could not find them in the manuscript.

Answer to reviewer’s comment: Thank you for your feedback. Sorry that you haven’t had an access to those but we did provide those with our submission. We are providing them again with this revised version at the end of the manuscript. Page 14 to 21

  1. In the results session, you only need to present the results from the included studies. You mixed too much information. For example, in line 214-216, this is something you need to write in discussion part.

Answer to reviewer’s comment: Thank you for your comment. Sections which include extra information have been removed. The results section now only presents data from included studies and we have removed that information from the results section

 Other comments

  1. Impact on routine immunization: You only write the rate of decline in vaccine coverage, but it is better to include the vaccination coverage before and during pandemic.

Answer to reviewer’s comment: In our review, we have attempted to report both pre and post pandemic data. However due to limited data, not all studies reported data previous data, but where this was available we have put it, please refer to table 2 on page 17. This has been identified as a limitation in: 376-385

  1. Line202-207: I cannot understand this part. You said that the impact was not just limited to HICs in first sentence, but you said the health care center observed no reductions in activities.

Answer to reviewer’s comment: Edited for clarity: line 226-228.

  1. a decline (57%) in vaccinations rates during March 1 - April 26, 2020 compared to the same period last year (March 1 - April 26, 2019). Inequality in vaccination coverage: This section actually does not address the inequality of the vaccine coverage.

Answer to reviewer’s comment: thanks. Line 238: Removed it for clarity

  1. Impact on mass immunization or SIAs: In this section, you just present the increase in polio cases. How did you know this is the effect of mass immunization or SIAs?

Answer to reviewer’s comment: Thank you for your comment. Polio currently only exists in Pakistan and Afghanistan and efforts to reduce and eliminate polio are conducted as part of mass immunisation programs and SIAs. You are right and we have been very conservative in showing this finding and never clearly relate that this impact was due the pandemic, but do in fact state the number of cases were higher before the pandemic.

Round 2

Reviewer 1 Report

The authors responded to the reviewer's comments, unfortunately one point was omitted and no answer was given. Therefore, I recommend the article to be revised by the authors.

Author Response

The authors responded to the reviewer's comments, unfortunately one point was omitted and no answer was given. Therefore, I recommend the article to be revised by the authors. Answer to reviewer’s comment: Thank you for bringing this to our notice, we had missed adding two references. We have now added these two missing references (doi: 10.3390/pathogens9060493; doi: 10.3390/pathogens9060501) in the discussion section as reference # 61 and 62.

Reviewer 4 Report

Thank you very much for your revision. Unfortunately, I could not see much progress from the previous version. Your main findings are not organized systematically, it is not easy to catch the important points from the current version of your manuscript. Please check PRISMA as well as the review comments for the previous rounds. 

Author Response

Thank you very much for your revision. Unfortunately, I could not see much progress from the previous version. Your main findings are not organized systematically, it is not easy to catch the important points from the current version of your manuscript. Please check PRISMA as well as the review comments for the previous rounds. 

Answer to reviewer’s comment: Thank you for your review and coments. We have now relooked into the previous comments and made considerable edits to the draft. We have tried to as clear and systematic in the results description. But it should be noted that though this is a systematic review but we have just performed a descriptive analysis due to the limited and varying number of studies and this limits the description of results as we have to describe the results individually. We hope that this verison would be better and acceptable.   

The PRISMA checklist is added as a supplementray file for clarity.

Also pasted below are the comments made in round 1 with our answers to those.

 Comments from Round 1

Editorial comment

MDPI would like to ensure accurate and correct communication of information around Covid-19. We would therefore like to note some common terms that might be used more accurately. Some terms for your consideration might include, for example: 'Originated' might be more accurate as 'reported'

'discovered' or 'identified'; 'COVID-19-infected people' might be more accurate as 'people/person with COVID-19' 'War on coronavirus' might be more accurate as 'protect/protection from COVID-19'.

Answer to an Editorial comment:  Thank you for your inputs. We have now replaced these, where applicable.

Reviewer 1

In the manuscript presented for review, Dr. Lassi and colleagues provided an important systematic review that clearly showed that vaccination coverage during the COVID-19 pandemic is significantly reduced, with other infectious diseases becoming more common.

The article is written carefully, but requires a few changes:

  1. the introduction is definitely too long and must be shortened,

Answer to reviewer’s comment: Thank you for your feedback. We have revisited the introduction section and have shortened it.

  1. the publication does not contain figures and tables, despite the fact that the authors cite them in the text of the article,

Answer to reviewer’s comment: Thank you for your feedback. Sorry that you haven’t had an access to those but we did provide those with our submission. We are now providing them again with this revised version from Page 14 to 21

  1. I can't find bias risk assessment in individual studies either. Regardless of the fact that the authors did not perform the mat-analysis, such an assessment must be included in the paper,

Answer to reviewer’s comment: Thank you for your feedback. Sorry that you haven’t had an access to the risk of bias assessment table. We are providing them again with this revised version as table 3 on page 21

  1. the search strategy should be supplemented with the number of results for at least one database.

Answer to reviewer’s comment: We have added the numbers to the Supplementary file 2

  1. Literature is extensive which is fine, please check the formatting.

Answer to reviewer’s comment: we have revisited the entire manuscript and formatted where required. 

Reviewer 2 The review presents an important question but needs considerable work before meriting publication to the standards expected of IJERPH.

  1. The methodology is not described in sufficient detail and needs considerable development. The exclusion criteria for studies needs to be properly described in a lot more detail.

Answer to reviewer’s comment: Thank you for your feedback. To ensure detailed description, changes have been made. For reference: Line: 103-121

  1. Diagrams are essential here within the methods to help communicate what studies were lacking which justified exclusion. A standard PRISMA diagram would be advised.

Answer to reviewer’s comment: Thank you for your feedback. Sorry that you haven’t had an access to the figures and tables, we are providing them again and at the end of the manuscript with fig 1 as PRISMA on page 21 

  1. A table of the papers which were able to be included with further detail on which disease/location/main findings/other classification criteria would be useful. This should be grouped by disease. 

Answer to reviewer’s comment: The table with the included studies is provided as table 1 and now we are putting this at the end of the manuscript on page 15-21

  1. The results need figures and tables summarising the information. The figures should be demonstrating a numerical output (i.e. decline in vaccination rate) which allows the reader to synthesise the information. This should be broken down into HMIC and LMIC where applicable or split into groups which reflect the discussion of the main findings.

Answer to reviewer’s comment: Sorry that you did not get access to the figures/tables. The tables have these information and we are now providing them again at the end of the manuscript (page 14 to 21) and specifically see table 2 (page 24).

  1. At this current presentation, it is almost impossible to follow as there is a lot of regurgitation of individual paper's main findings with insufficient binding narrative which allows the reader to draw interesting insights or concrete take home messages.

Answer to reviewer’s comment: thank you for this comment, we have now revisited the manuscript and edited to give it a flow for readers.

Reviewer 3

Zohra S Lassi et al proposed a systematic review of immunization programs in different countries.

  1. Comment 1: Immunization programs are varied in different countries based on different policies. There is little evidence in this study to show the impact of COVID-19 or other factors that determine the differences in different countries.

Answer to reviewer’s comment: In this study, we attempt to study the pre and during/post COVID-19 impact on immunization programs across the globe. We do not compare the difference in countries but merely attempt to highlight the impact of COVID-19 on preexisting programs and their policies. This would have been a very useful information, but with the current limited data – this would be hard to infer. For clarity we have added this under limitations on page 14, lines: 376 - 385

  1. Comment 2: The cost of vaccines is very important for parents to decide what vaccines do they children will be taken in many circumstances. If this study only report observational studies, it is not enough.

Answer to reviewer’s comment: The objective of this systematic review was to review the uptake and coverage that has been affected because of COVID pandemic by comparing the coverage before and during/post COVID times. This could have been an important factor but was not found in the studies included.

  1. Comment 3: Figures and Tables are mentioned in the manuscript, but I could not find any in the manuscript and the supplementary.

Answer to reviewer’s comment: Thank you for your feedback. Sorry that you haven’t had an access to those but we did provide those with our submission. We are providing them again with this revised version at the end of the manuscript from page 14 to 21 

  1. Comment 4: And statistical methods/results that merge data from different studies are lacking.

Answer to reviewer’s comment: Due to the heterogeneity in data, it was not possible to merge data and execute any statistical analysis in our review. This has been reported on page 3, lines 141-144.

Reviewer 4

This manuscript highlights the important public health issue during COVID-19 pandemic. A lot of study focuses on COVID-19 and sometimes the impact of pandemic on other health care issues tend to be fallen behind. I think this manuscript is worth to be published, but it required extended revision. Please check the PRISMA checklist again.

  1. Introduction

Please specify the objectives. (PICOS)

For example, you need to address your target population (e.g. LMIC?)

Answer to reviewer’s comment: Thank you for your comment. We have edited this under both introduction and methods: 86-96 and 107-114

  1. Please specify why you need to conduct this study.

Answer to reviewer’s comment: Thank you for your feedback. We have now added this at the end of the background (line 94-96) “The findings from this study would provide insights for the impact of pandemic on existing vaccination strategies and hence help inform policy for effective planning for future pandemics and other crisis.”

  1. What is your current knowledge about the impact of COVID-19 pandemic on vaccination program from the previous study?

Answer to reviewer’s comment: To the best of our knowledge, this specific topic has not been reviewed before. In our review, we have attempted to explore all studies that have reported the impact on vaccination programs.

Methods

  1. Please specify inclusion and exclusion criteria.

Answer to reviewer’s comment: Thank you for your feedback. Required changes can be found at line 107 to 114

  1. In outcome measure, you include inequality in immunization coverage. How do you address the inequality?

Answer to reviewer’s comment: In order to avoid any confusion, we have removed this as an outcome measure. Line 145-146, page 4

Results

Overall

  1. You write Figure and Tables in the text. But I could not find them in the manuscript.

Answer to reviewer’s comment: Thank you for your feedback. Sorry that you haven’t had an access to those but we did provide those with our submission. We are providing them again with this revised version at the end of the manuscript. Page 14 to 21

  1. In the results session, you only need to present the results from the included studies. You mixed too much information. For example, in line 214-216, this is something you need to write in discussion part.

Answer to reviewer’s comment: Thank you for your comment. Sections which include extra information have been removed. The results section now only presents data from included studies and we have removed that information from the results section

 Other comments

  1. Impact on routine immunization: You only write the rate of decline in vaccine coverage, but it is better to include the vaccination coverage before and during pandemic.

Answer to reviewer’s comment: In our review, we have attempted to report both pre and post pandemic data. However due to limited data, not all studies reported data previous data, but where this was available we have put it, please refer to table 2 on page 17. This has been identified as a limitation in: 376-385

  1. Line202-207: I cannot understand this part. You said that the impact was not just limited to HICs in first sentence, but you said the health care center observed no reductions in activities.

Answer to reviewer’s comment: Edited for clarity: line 226-228.

  1. a decline (57%) in vaccinations rates during March 1 - April 26, 2020 compared to the same period last year (March 1 - April 26, 2019). Inequality in vaccination coverage: This section actually does not address the inequality of the vaccine coverage.

Answer to reviewer’s comment: thanks. Line 238: Removed it for clarity

  1. Impact on mass immunization or SIAs: In this section, you just present the increase in polio cases. How did you know this is the effect of mass immunization or SIAs?

Answer to reviewer’s comment: Thank you for your comment. Polio currently only exists in Pakistan and Afghanistan and efforts to reduce and eliminate polio are conducted as part of mass immunisation programs and SIAs. You are right and we have been very conservative in showing this finding and never clearly relate that this impact was due the pandemic, but do in fact state the number of cases were higher before the pandemic.

Round 3

Reviewer 1 Report

The authors have satisfactorily responded to all my questions and made the necessary changes to the manuscript.

Reviewer 4 Report

I think this manuscript is ready for the submission.